# Transcriptome Analysis of Eggplant under Salt Stress: AP2/ERF Transcription Factor SmERF1 Acts as a Positive Regulator of Salt Stress

**DOI:** 10.3390/plants11172205

**Published:** 2022-08-25

**Authors:** Lei Shen, Enpeng Zhao, Ruie Liu, Xu Yang

**Affiliations:** 1College of Horticulture and Plant Protection, Yangzhou University, Yangzhou 225009, China; 2Shanghai Center for Plant Stress Biology, National Key Laboratory of Plant Molecular Genetics, Center for Excellence in Molecular Plant Sciences, Chinese Academy of Sciences, Shanghai 201600, China

**Keywords:** eggplant, salt stress, ERF transcription factor, *SmERF1*, *Solanum melongena*

## Abstract

Salt stress, a type of abiotic stress, impedes plant growth and development and strongly reduces crop yield. The molecular mechanisms underlying plant responses to salt stress remain largely unclear. To characterize the enriched pathways and genes that were affected during salt treatment, we performed mRNA sequencing (mRNA-seq) in eggplant roots and identified 8509 differentially expressed genes (DEGs) between the mock and 24 h under salt stress. Among these DEGs, we found that the AP2/ERF transcription factor family member *SmERF1* belongs to the plant–pathogen interaction pathway, which was significantly upregulated by salt stress. We found that SmERF1 localizes in the nuclei with transcriptional activity. The results of the virus-induced gene silencing assay showed that *SmERF1* silencing markedly enhanced the susceptibility of plants to salt stress, significantly downregulated the transcript expression levels of salt stress defense-related marker genes (*9-cis-epoxycarotenoid dioxygenase* [*SmNCED1*, *SmNCED2*], *Dehydrin* [*SmDHN1*], and *Dehydrin* (*SmDHNX1*), and reduced the activity of superoxide dismutase and catalase. Silencing *SmERF1* promoted the generation of H_2_O_2_ and proline. In addition, the transient overexpression of *SmERF1* triggered intense cell death in eggplant leaves, as assessed by the darker diaminobenzidine and trypan blue staining. These findings suggest that SmERF1 acts as a positive regulator of eggplant response to salt stress. Hence, our results suggest that AP2/ERF transcription factors play a vital role in the response to salt stress.

## 1. Introduction

With continuous changes in the ecological environment, the area of saline-alkali land worldwide is increasing every year, becoming a global, burning issue [1]. Secondary salinization caused by the excessive use of fertilizers and irrational soil cultivation seriously affects the growth, development, yield, and quality of crops [2,3]. Salt stress affects water absorption, leading to cell dehydration and osmotic stress [4]. The excessive accumulation of salt ions caused by water transpiration triggers ionic stress in plant cells, which affects plant morphology and physiological and biochemical processes and ultimately limits plant growth and development [5]. High salinity also affects plant photosynthesis by regulating gene expression and stomatal closure, ultimately leading to a reduction in crop productivity due to decreased carbon dioxide absorption [6].

mRNA sequencing (mRNA-seq) based on transcriptome profiling has become an efficient, routine technology for the signaling pathway analysis of transcriptional regulation in plant responses to salt stress [7]. Under salt stress, many defense-related genes are activated by transcriptional reprogramming in plants, a process in which transcription factors play vital roles [8,9]. Apetala2/ethylene-responsive factor (AP2/ERF) is a major plant-specific family of transcription factors comprised of four main functional domains: the nuclear localization signal region, DNA-binding domain, transcriptional regulatory domain, and oligomerization site [10]. AP2/ERF transcription factor superfamily members have one or more typical AP2/ERF domains composed of 60–70 highly conserved amino acids that recognize and bind DNA [11]. According to sequence similarity and the number of AP2/ERF domains, the AP2/ERF transcription factor superfamily is divided into three main families: AP2, ERF, and RAV (related to ABI3/VP1). AP2 family members contain two conserved AP2 domains, RAV family members harbor an AP2 domain and a B3 domain, and ERF family members have an AP2 domain and a WLG motif [12]. The ERF transcription factor family can be divided into two subfamilies: dehydration-responsive element-binding protein (DREB) and ERF [13]. The sequences of the AP2/ERF-conserved domains are relatively similar, and they differ in the type of *cis*-elements within the target gene promoter bound by subtribe members of the AP2/ERF transcription factor [14,15]. DREB proteins recognize and bind to the drought-responsive *cis*-element harboring the A/GCCGAC core sequence [16], while ERF proteins bind to the GCC box (AGCCGCC) *cis*-element [17]. ERF transcription factors are mostly involved in ethylene and abscisic acid (ABA) signaling pathways and function in plant responses to salt [18,19,20], cold [21,22,23], heat [24,25], drought [26,27,28,29], and pathogen attacks [30,31,32]. These transcription factors regulate the expression of the downstream stress-response genes by binding to specific *cis*-elements within the promoters [33,34]. However, the roles and mechanisms of most ERF transcription factors in eggplant responses to salt stress remain largely unknown.

Abscisic acid (ABA) acts as a phytohormone and plays a vital role in plant responses to abiotic stress including salt and drought stress [35,36,37]. Salt or drought stress can cause the endogenous plant ABA level to rise by 40-fold [38]. The activation of ABA biosynthetic enzymes plays a vital role in the plant against salt and drought stresses. During ABA biosynthesis, 9-cis-epoxycarotenoid dioxygenases (NCEDs) can catalyze 9-cis-epoxycarotenoids, the cleavage of 9-cis-epoxycarotenoids, to xanthoxin [39]. In *Arabidopsis thaliana*, *NCED3* acts as a key regulation factor in ABA biosynthesis in plant responses to water deficit and salt stress [40,41]. The transcription factor *NGATHA1* can regulate the *NCED3* expression level in the *Arabidopsis thaliana* response to salt stress [42]. 

Eggplant is an agriculturally vital solanaceous vegetable cultivated worldwide [43]. Salt stress critically affects eggplant yield; however, the mechanisms underlying the salt stress response of eggplant remain unclear. In this study, we aimed to investigate the response mechanisms of eggplant to salt stress and identify the key genes involved in eggplant responses to salt stress. We performed an RNA sequencing assay to screen the differentially expressed genes (DEGs) in the eggplant roots under salt stress and identified the different metabolic pathways involved in eggplant responses to salt stress by gene ontology (GO) annotation and Kyoto Encyclopedia of Genes and Genomes (KEGG) analysis. We selected and characterized an AP2/ERF transcription factor family member, *SmERF1*, which acts as a positive regulator of the eggplant response to salt stress.

## 2. Results

### 2.1. Sequencing Data Summary and DEGs Analysis

Transcriptional assays in the eggplant inbred line ML41 were performed to screen potential target genes involved in the response to salt stress. The roots of ML41 treated with 150 mM NaCl solution were harvested at 0 and 24 h post-treatment for mRNA-seq analysis [7]. Illumina high-throughput sequencing yielded 281.9 million clean reads, corresponding to an average of 47 million sequence reads per sample. Approximately 91.53% of the paired-end clean reads were aligned to the *Solanum melongena* reference genome (http://eggplant-hq.cn/Eggplant/home/index, accessed on 20 July 2022) using Hisat2 [44,45] (Appendix A). The spatiotemporal expression patterns of the samples were investigated using the principal component analysis (PCA). The samples from 0 and 24 h post-treatment formed independent clusters (Figure 1a). Compared to the samples at 0 h post-treatment, 8509 DEGs, comprising 4878 up-regulated and 3631 down-regulated genes, were identified in the samples at 24 h post-treatment. The identified DGEs between the 0 and 24 h post-treatment groups are visualized with the Volcano plot (Figure 1b). The heat map shows that a mass of DEGs was identified in the samples at 24 h post-treatment compared with the samples at 0 h post-treatment (Figure 1c). The results indicated that salt stress led to comprehensive transcriptome changes in the eggplant root cells.

### 2.2. GO Analysis of DEGs

The functional classification of the DEGs induced by salt treatment was investigated using GO enrichment analysis. The top 20 enriched GO terms included biological processes, molecular functions, and cellular components (Figure 2). The upregulated DEGs were significantly assigned to 127 pathways in biological processes, 82 pathways in molecular functions, and 67 pathways in cellular components; the downregulated DEGs were assigned to 215 pathways related to biological processes, 96 pathways in molecular functions, and 34 pathways in cellular components (Appendix A).

Within biological processes, upregulated DEGs were significantly enriched in 127 pathways, including the response to oxygen-containing compounds (GO:1901700) related to ABA responsiveness [46], the response to acid chemicals (GO:0001101) such as pyruvic acid, which is related to the process of ABA biosynthesis [47], and the response to water (GO:0009415) (Appendix A). This indicated the role of ABA in plant responses to salt treatment and suggested that the expression of the genes upregulated by salt may involve in these pathways. 

### 2.3. KEGG Analysis of DEGs

KEGG analyses were conducted to further understand the function of the DEGs associated with the salt response. The 14 up-enrichment pathways and 18 down-enrichment pathways were enriched by salt stress treatment in eggplant roots, respectively (Appendix A). The up-regulated genes were significantly enriched in the categories “plant-pathogen interaction” and “plant hormone signal transduction” (Figure 3a), whereas the down-regulated genes were enriched for the categories “metabolic pathways”, “biosynthesis of secondary metabolites”, “starch and sucrose metabolism”, and “pentose and glucuronate interconversions” (Figure 3b). These results indicated that salt stress induces multiple biological processes in eggplants at the whole-genome level. 

The results of the KEGG analyses showed that salt stress treatment upregulates the transcript expression levels of genes related to plant–pathogen interaction and plant hormone signal transduction (Figure 3a). Therefore, we selected eight plant–pathogen interaction-related genes (Appendix A), namely, *SmMAPKKK1*, *SmWRKY20*, *SmRIN1*, *SmERF1*, *SmHSP90*-5, *SmCNGC17*, *SmCDPK11*, and *SmCDPK32*, and five ABA signal transduction-related genes (Appendix A), including the environmental stress-inducible protein *TAS14* (*SmTAS14*), a late embryogenesis abundant protein family member *Dehydrin*
*1* (*SmDHN1*) [48], *Dehydrin Xero 1* (*SmDHNX1*), which is regulated by ABA signaling [49], *abscisic acid 8′-hydroxylase 3* (*SmABA8′HO3*) [25], and *glutathione S-transferase U10* (*SmGSTU10*), to detect their transcript expression levels in eggplant roots under salt stress by a quantitative real-time PCR (qPCR) assay. Four biological replications (one plant root as a biological repeat) were used for the detection of above-mentioned genes’ transcript expression levels treated with salt stress at 0, 3, 6, 12, 24, and 48 h post treatment. The transcript expression levels of these genes were significantly upregulated by the salt stress treatment (Figure 4), which is consistent with the mRNA-seq data. Similarly, salt stress significantly upregulated the transcript expression levels of the salt stress defense-related marker genes *9-cis-epoxycarotenoid dioxygenase*
*1* (*SmNCED1*) and *SmNCED2* [40,50] (Figure 5). 

### 2.4. Sequence Analysis of SmERF1

The KEGG analysis showed that *SmERF1* belongs to the plant–pathogen interaction pathway, and its relative transcript expression level was significantly upregulated by salt stress treatment. The fold of the transcript expression level of *SmERF1* up-regulated at 48 h post-salt stress treatment was higher than the other seven selected genes. This suggests that *SmERF1* may play a role in eggplant responses to salt stress or pathogen attacks. To verify the function of *SmERF1* in the eggplant response to salt stress, we first analyzed the sequence of *SmERF1*. The full-length open reading frame (ORF) sequence of *SmERF1* was obtained by searching the Eggplant Genome Database (http://eggplant-hq.cn/Eggplant/home/index, accessed on 20 July 2022) using the gene ID (Smechr0500241.1). The ORF of *SmERF1* (711 bp), which encodes a 26 kDa protein, was amplified by PCR and cloned into the plant overexpressing vector pBinGFP2 (containing a green fluorescence protein [GFP] tag). Within the deduced amino acid sequence of SmERF1, there was a conserved APETALA2 (AP2) domain, which is necessary and sufficient for ERF transcription factors to bind to the GCC-box [51] (Appendix A). The multiple sequence alignment showed that *SmERF1* shared 86, 83, 70, 70, 60, 55, 53, 53, and 51% sequence identity with its orthologs in *Solanum tuberosum ERF1* (XP_006365341.2), *Nicotiana tabacum ERF1* (XP_016484768.1), *Solanum lycopersicum ERFA.2* (NP_001316388.2), *Actinidia chinensis* var. *chinensis ERF2* (PSS33216.1), *Capsicum annuum ERF1* (XP_047262199.1), *Gossypium hirsutum ERF2* (XP_016677202.1), *Zea mays ERF2* (XP_008646008.1), *Arabidopsis thaliana ERF1A* (CAB45963.1), and *Oryza sativa ERF1* (CAC39058.1), respectively (Appendix A). Phylogenetic analysis of the *SmERF1* amino acid sequence and its orthologs revealed *SmERF1* to be most closely related to *Solanum tuberosum ERF1* (Appendix A). In addition, we analyzed *cis*-elements within the promoter sequence of *SmERF1* using PlantCARE (http://bioinformatics.psb.ugent.be/webtools/plantcare/html/, accessed on 20 July 2022) website tools and identified the stress- or pathogen-responsive *cis*-elements: the CGTCA motif involved in methyl jasmonate (MeJA) responsiveness, the TCA-element involved in salicylic acid (SA) responsiveness, the ABA-responsive element (ABRE), the AAGAA motif related to ABA responsiveness, the TC-rich repeats involved in defense and stress responsiveness, the stress response element (STRE), and the MYB transcription factor binding site (Appendix A). These results implied that *SmERF1* is likely involved in the eggplant response to abiotic stress or pathogenic infection.

### 2.5. Subcellular Localization of SmERF1

We analyzed the subcellular localization of SmERF1 in the epidermal cells of *N. benthamiana* leaves. The full-length ORF of *SmERF1* was cloned into the destination vector pBinGFP2 to generate the recombinant vector pBinGFP2-*SmERF1* (*35S:SmERF1-GFP*); the empty vector pBinGFP2 (*35S:GFP*) was used as a control (Appendix A). Cells of the *Agrobacterium tumefaciens* strain GV3101 carrying *35S:SmERF1-GFP* (containing a GFP tag), *35S:H2B-RFP* (containing a red fluorescence protein tag and acting as a marker of nuclear localization), or *35S:GFP* constructs were cultivated in liquid Luria–Bertani (LB) medium overnight, and the bacterial fluid was infiltrated into *N. benthamiana* leaves. Green and red fluorescence signals in the epidermal cells of *N. benthamiana* leaves were detected at 48 h post-infiltration. Both green fluorescence signals of *35S:SmERF1-GFP* and red fluorescence signals of *35S:H2B-RFP* were observed in the nuclei, whereas the green fluorescence signals of *35S:GFP* appeared throughout the cell (Appendix A).

### 2.6. Transcriptional Activation Activity Analysis

Accumulating evidence has demonstrated that ERF transcription factors play a vital role in plant responses to biotic or abiotic stresses by activating the expression of downstream target genes [31,52,53,54,55]. To investigate the *SmERF1* activity in transcriptional activation, we performed a transcriptional activation activity assay using the yeast GAL4 system. The full-length ORF of *SmERF1* was cloned into the bait vector pGBKT7 to generate the recombinant vector pGBKT7-*SmERF1* (Figure 6a). Yeast cells (AH109) containing the pGBKT7-*SmERF1* construct and those with pGBKT7 formed colonies on the solid yeast peptone dextrose adenine (YPDA) medium lacking tryptophan (Trp). We further inoculated the yeast cells with pGBKT7-*SmERF1* or pGBKT7 on a solid YPDA medium containing X-α-Gal but lacking both Trp and histidine (His). The yeast cells carrying pGBKT7-*SmERF1* constructs grew well and formed blue bacterial colonies. In contrast, the yeast cells carrying the pGBKT7 vector did not survive on the solid YPDA medium supplemented with X-α-Gal and lacking Trp and His (Figure 6b), suggesting that SmERF1 has transcriptional activation activity.

### 2.7. *SmERF1* Silencing Enhanced the Susceptibility of Eggplant to Salt Stress

For the silencing of *SmERF1*, *Agrobacterium tumefaciens* GV3101 cells transformed with the pTRV1 vector were mixed with the cells carrying pTRV2:*SmERF1*, pTRV2:*00*, or pTRV2:*SmPDS* constructs, and the bacterial fluid was infiltrated into the cotyledon of 2–4 leaf-stage eggplant seedlings. When photobleaching occurred in the eggplant plant leaves infiltrated with *Agrobacterium tumefaciens* GV3101 cells contenting TRV:*SmPDS* constructs, four biological repeats (one plant root as a biological repeat) of the control (TRV:*00*) or *SmERF1*-silenced (TRV:*SmERF1*) plants were used to detect the silencing efficiency of *SmERF1* using qPCR in the conditions of salt stress treatment. The relative transcript expression levels of *SmERF1* in *SmERF1*-silenced (TRV:*SmERF1*) eggplant roots were significantly reduced by approximately 85% compared with those in the control plants (Figure 7a). The TRV:*SmERF1* and TRV:*00* plants were treated with 150 mM NaCl solution to assess their tolerance to salt stress. The TRV:*SmERF1* plants displayed more serious wilt symptoms and lower survival rates than the TRV:*00* plants at 2 days post-treatment (Figure 7b,c). Compared with the control plants, *SmERF1* silencing significantly reduced the relative transcript expression levels of the defense-related marker genes *SmNCED1*, *SmNCED2*, *SmDHN1*, and *SmDHNX1*, which were respectively reduced by approximately 81%, 86%, 93%, and 92%, in *SmERF1*-silenced eggplant roots (Figure 7d). We also monitored the enzyme activity of superoxide dismutase (SOD) and catalase (CAT), as well as the content of H_2_O_2_ and proline, and four biological repeats at each time point were used to detect the above-mentioned physiological indexes and found that *SmERF1* silencing significantly reduced the accumulation of H_2_O_2_ and proline and the activity of SOD and CAT compared to the control plants in the 150 mM NaCl solution treatment (Figure 7e). These data suggest that *SmERF1* acts as a positive regulator of eggplant responses to salt stress.

### 2.8. The Transient Overexpression of *SmERF1* Triggers Cell Death in Eggplant Leaves

The analysis of the mRNA-seq data revealed that *SmERF1* belongs to the plant immune response pathway. To investigate the role of SmERF1 in the eggplant immune response, we transiently overexpressed 35S:*SmERF1*-GFP or 35S:*00* (empty vector, EV) in eggplant leaves. We first detected the expression of *SmERF1* in eggplant leaves (four biological repeats) 48 h post-infiltration by a qPCR assay, thus confirming its successful expression (Figure 8a). In addition, the transient overexpression of *SmERF1* accumulated more H_2_O_2_, as indicated by the diaminobenzidine (DAB) staining and the darker trypan blue staining of the eggplant leaves with 35S:*SmERF1* transient overexpression compared to the leaves with EV transient overexpression at 3 days post-infiltration (Figure 8b). These data indicate that *SmERF1* may function in the eggplant immune response.

## 3. Discussion

Salt stress is one of the most important abiotic stressors in eggplants. It reduces the yield and quality of eggplant by affecting physiological and biochemical processes such as water transport, cell osmotic pressure, and photosynthesis [4,5]. A better understanding of the underlying molecular mechanisms of eggplant response to salt stress will aid the breeding of varieties with improved salt tolerance. Through the mRNA-seq assay, we identified the DEGs in eggplant roots treated with salt stress and investigated the markedly enriched biological processes and pathways by GO and KEGG analyses. Accumulating evidence has demonstrated that AP2/ERF transcription factors play a vital role in plant resistance to salt stress [56,57,58,59]. An AP2/ERF transcription factor superfamily member, *SmERF1*, belonging to the plant–pathogen interaction pathway, was selected and was confirmed to act as a positive regulator in eggplant responses to salt stress. 

*SmERF1* (711 bp) encoded a 26 kDa protein and shared the highest sequence identity (86%) and the closest affinity with *Solanum tuberosum ERF1* (Appendix A). The promoter of *SmERF1* harbored stress− or pathogen-responsive *cis*−elements, suggesting that *SmERF1* may play a role in eggplant responses to biotic or abiotic stresses (Appendix A). Our data showed that the transcript expression level of *SmERF1* in eggplant roots was markedly upregulated by salt stress treatment (Figure 4a). SmERF1 localizes to the nucleus and activates the transcriptional activation (Figure 6 and Appendix A). In the VIGS assay, *SmERF1* silencing significantly enhanced the susceptibility of the plants to salt stress (Figure 7b) and down−regulated the transcript expression levels of the salt stress defense−related marker genes *SmNCED1* [40,50], *SmNCED2*, *SmDHN1* [48], and *SmDHNX1* (Figure 7d). These genes contribute to the plant’s response to salt stress via the ABA signaling pathway, which is accompanied by a reduction in SOD and CAT enzyme activity and an increase in H_2_O_2_ and proline content (Figure 7e). These data indicate that *SmERF1* positively regulates eggplant responses to salt stress and is closely related to ABA signaling networks. However, it remains unknown how *SmERF1* affects the transcript expression levels of the above-mentioned salt stress defense-related marker genes. One possible explanation is that SmERF1 binds to the dehydration-responsive element (DRE) core *cis*-element within the *SmDHN1* promoter, regulating eggplant tolerance to salt stress (Figure 7d and Figure 9). KEGG analyses revealed that *SmERF1* belongs to the plant–pathogen interaction pathway (Appendix A), and intense cell death was triggered in eggplant leaves by the transient overexpression of *SmERF1* (Figure 8b). The AP2/ERF transcription factors *CaAP2*/*ERF064* and *CaAIL1* can trigger reactive oxygen species (ROS) burst, leading to intense cell death in pepper and *N. benthamiana* leaves, as they mediate the immunity of these species [60,61]. Therefore, it can be speculated that *SmERF1* positively regulates the plant immune response by mediating ROS signaling networks. In conclusion, our data suggest that *SmERF1* acts as a positive regulator of salt stress in eggplants.

## 4. Materials and Methods

### 4.1. Plant Material and Salt Treatment

The seeds of the inbred eggplant line ML41 were germinated by wrapping them in moistened gauze and incubating them at 25 °C for 1 week. For mRNA-seq analysis, the roots of 4–6 leaf-stage seedlings with a uniform size were soaked in Hoagland nutrient solution, and the seedlings were grown at 25 °C under a 16 h light/8 h dark photoperiod with 800 μmol m^−2^ s^−2^ illumination. For the virus-induced gene silencing and subcellular localization assay, the ML41 and *N. benthamiana* seedlings were grown in plastic flowerpots filled with sterilized soil in an illumination incubator at 25 °C under a 16 h light/8 h dark photoperiod with 800 μmol m^−2^ s^−2^ illumination.

For the salt treatment, the roots of 4–6 leaf-stage eggplant seedlings were soaked in 150 mM NaCl solution for 24 h; ddH_2_O treatment served as a control. Three different samples were used as biological replicates for the salt treatment. The eggplant roots were washed with ddH_2_O, frozen in liquid nitrogen, and stored at −80 °C for mRNA-seq analysis.

### 4.2. Total RNA Extraction, cDNA Library Construction, and Illumina Sequencing

The total RNA was extracted from the NaCl-treated and ddH_2_O-treated roots of eggplant seedlings using a FastPure Plant Total RNA Isolation Kit (Vazyme, Nanjing, China), according to the manufacturer’s instructions. RNA purity and concentration were examined using a NanoDrop 2000 spectrophotometer (Thermo Fisher Scientific, Waltham, MA, USA), and RNA integrity and quantity were measured using an Agilent 2100/4200 system (Agilent Technologies, Santa Clara, CA, USA).

cDNA sequencing libraries were constructed using a TruSeq RNA Sample Preparation kit (Illumina, San Diego, CA, USA), following the manufacturer’s instructions. mRNA was purified from the total RNA using poly-T and then fragmented into 300–350 bp fragments. First-strand cDNA was reverse-transcribed using fragmented RNA and dNTPs, and second-strand cDNA was synthesized subsequently. The sequencing adaptors were ligated to cDNA, and the library fragments were purified. The template was enriched by PCR, and the PCR product was purified to obtain the final library.

Paired-end reads of 150 bp were generated by sequencing on an Illumina HiSeq 2000 system after library preparation and the pooling of different samples. 

### 4.3. mRNA-Seq Read Processing and Assembly

Raw data (raw reads) in FASTQ format were generated using an Illumina HiSeq X-ten genome analyzer. Fast Q was used to analyze the raw reads and assess the base quality [62]. Clean data (clean reads) were obtained by removing the reads harboring adaptor sequences and low-quality reads with more than 20% nucleotides with Qphred ≤ 5. The Q20, Q30, and GC contents of the clean data were calculated, and the SILVA database (https://www.arb-silva.de/, accessed on 20 July 2022) was used to map the clean reads. Paired-end clean reads were aligned to eggplant reference genome sequences using HISAT2 with the default settings [44,45].

### 4.4. Sequencing Data Processing and Analysis 

The fragments per kilobase per million mapped reads (FPKM) value was used to assess the transcript expression levels of the genes. The FPKM value was measured using EdgeR to identify DEGs [63]. Genes with a |log2 (Fold Change)| > 1 & a q value < 0.05 were assigned as significantly differently expressed. Hierarchical cluster analysis was performed on the DEGs by using the heatmap package in R language. To investigate the biological function of the DEGs, the pathways enriched under salt stress treatment were identified using GO and KEGG analyses, as described previously [64,65]. The q value < 0.05 was assigned as significant enrichment.

### 4.5. Vector Construction

For the subcellular localization and transcriptional activation of SmERF1, the full-length ORF of *SmERF1* was amplified by PCR and cloned into the destination vector pBinGFP2 or pGBKT7 using a ClonExpress II One Step Cloning Kit (Vazyme), according to the manufacturer’s instructions. 

For the VIGS assay, a specific 300 bp DNA fragment from the *SmERF1* full-length ORF was amplified and cloned into the entry vector pDONR207 by a BP recombination reaction and then transferred into the destination vector pTRV2 by LR recombination, according to the manufacturer’s instructions.

### 4.6. Agrobacterium tumefaciens Cultivation and Infiltration and Subcellular Localization Analysis of SmERF1

The recombinant vector pBinGFP2-*SmERF1* was transformed into competent cells of the *Agrobacterium tumefaciens* strain GV3101. Positive GV3101 cells carrying *35S:SmERF1-GFP*, *35S:GFP*, or *35S:H2B-RFP* constructs were cultivated in liquid LB medium containing 50 mg/mL kanamycin and 50 mg/mL rifampicin at 28 °C and 200 rpm overnight. The bacterial culture was centrifuged at 28 °C and 6000 rpm for 10 min and resuspended in an infiltration medium (10 mM MES, 10 mM MgCl_2_, 200 mM acetosyringone, pH = 5.4) to adjust the OD_600_ to 0.8. The bacterial fluid of GV3101 cells harboring 35S:H2B-RFP constructs was mixed with cells carrying *35S:SmERF1-GFP* or *35S:GFP* constructs at a 1:1 ratio. The mixed bacterial fluid was infiltrated into the leaves of *N. benthamiana* using a disposable sterile syringe without a needle. Fluorescence signals were observed using a laser scanning confocal microscope (TCS SP8; Leica Microsystems, Wetzlar, Germany).

### 4.7. VIGS Assay

The VIGS assay was performed as previously described [66]. The function of *SmERF1* in eggplant responses to salt stress was verified by conducting a VIGS assay. A specific 300 bp DNA fragment from the full-length ORF of *SmERF1* was amplified by PCR, cloned into the entry vector pDONR207, and transferred into the destination vector pTRV2. *Agrobacterium tumefaciens* GV3101 cells carrying the pTRV1 vector were mixed with cells containing pTRV2:*SmERF1*, pTRV2:*00*, or pTRV2:*SmPDS* constructs at a 1:1 ratio. The mixtures were slowly incubated at 28 °C and 60 rpm for 3 h and injected into the cotyledon of 2–4 leaf-stage eggplant seedlings. The infiltrated seedlings were placed at 25 °C without light for 24 h and then transferred to an illumination incubator at 25 °C under a 16 h light/8 h dark photoperiod.

### 4.8. qPCR Assay

The qPCR assay was performed as previously described to analyze the transcript expression levels of the selected genes [66,67], and four biological repeats were used to detect the transcript expression levels of the target genes. The primer pairs used in this study are listed in Appendix A.

### 4.9. Physiological Index Measurement

The contents of proline and H_2_O_2_ and the enzyme activities of CAT and SOD were measured as previously described [68].

### 4.10. Histochemical Staining

Histochemical staining with DAB and trypan blue was performed as previously described [69].

## 5. Conclusions

In this study, we obtained the DEGs in the roots of eggplant plants treated with salt stress and investigated the biological processes and pathways significantly enriched by salt stress treatment. Eight genes related to the plant–pathogen interaction pathway and five genes associated with the ABA signal transduction pathway were selected, and the transcript expression levels of these genes were notably upregulated in the eggplant roots treated with salt stress. We further confirmed the positive function of *SmERF1*, which belongs to the plant–pathogen interaction pathway, in eggplant responses to salt stress and found that the transient overexpression of *SmERF1* triggered intense cell death in eggplant leaves. These results indicate that *SmERF1* acts as a positive regulator of salt stress in eggplant. In summary, our study provides the first report on the role of the ERF transcription factor in eggplant responses to salt stress and will help to further elucidate the mechanisms of *SmERF1*-regulated stress responses in eggplant. The results presented here lay the foundation for the effective genetic improvement of salt stress tolerance in eggplant.

## Figures and Tables

**Figure 1 plants-11-02205-f001:**
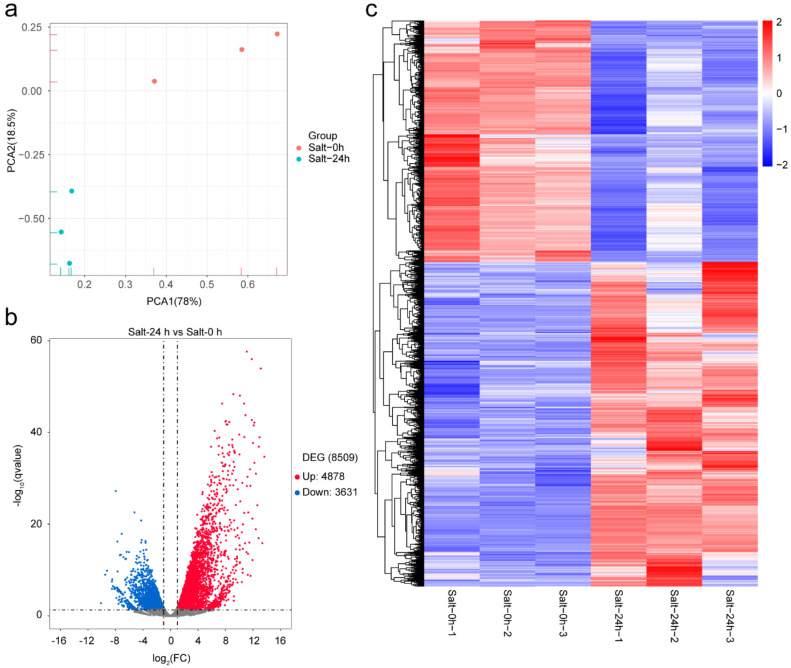
Spatiotemporal gene expression patterns under salt stress treatment. (**a**) Principal component analysis (PCA) of the samples at 0 and 24 h post−treatment based on the normalized expression levels of differentially expressed genes (DEGs). (**b**) Volcano plot of the differential expression levels of up-regulated and down−regulated DEGs in eggplant roots under salt stress. (**c**) Heat map of correlations between the transcript expression levels of DEGs in eggplant roots treated with salt stress. FPKM values were used for the hierarchical cluster analysis after the normalization of FPKM values with Z-score.

**Figure 2 plants-11-02205-f002:**
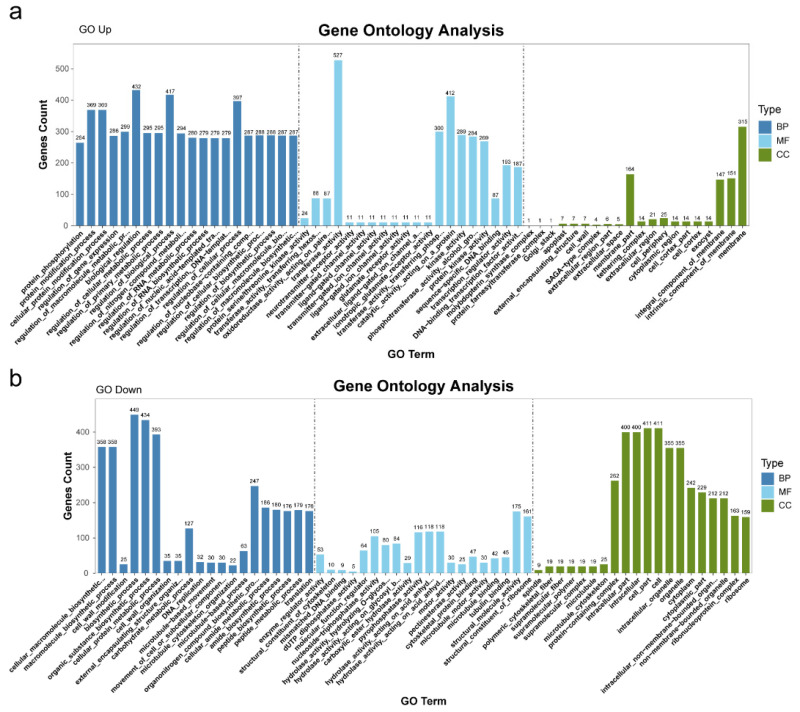
Functional classification of DEGs in eggplant roots under salt stress based on gene ontology (GO) analysis. GO analysis of up-regulated (**a**) and down-regulated (**b**) DEGs of the top 20 terms of biological processes (BP), molecular functions (MF), and cellular components (CC).

**Figure 3 plants-11-02205-f003:**
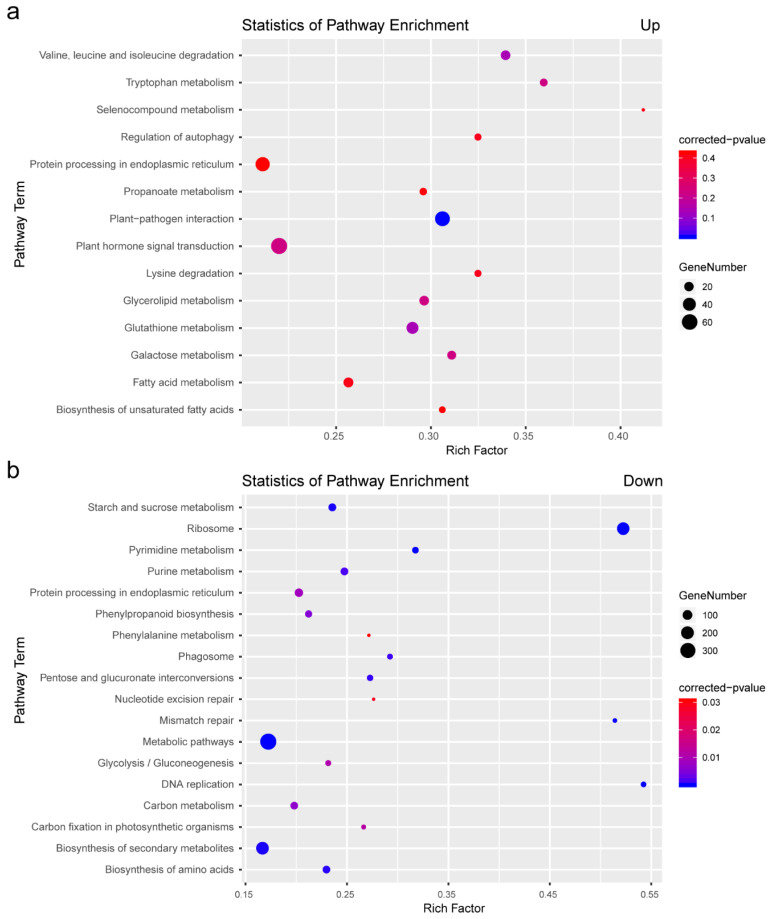
KEGG analysis of DEGs in eggplant roots under salt stress. KEGG analysis of the significantly up-regulated (**a**) and down-regulated (**b**) DEGs of the enriched pathways.

**Figure 4 plants-11-02205-f004:**
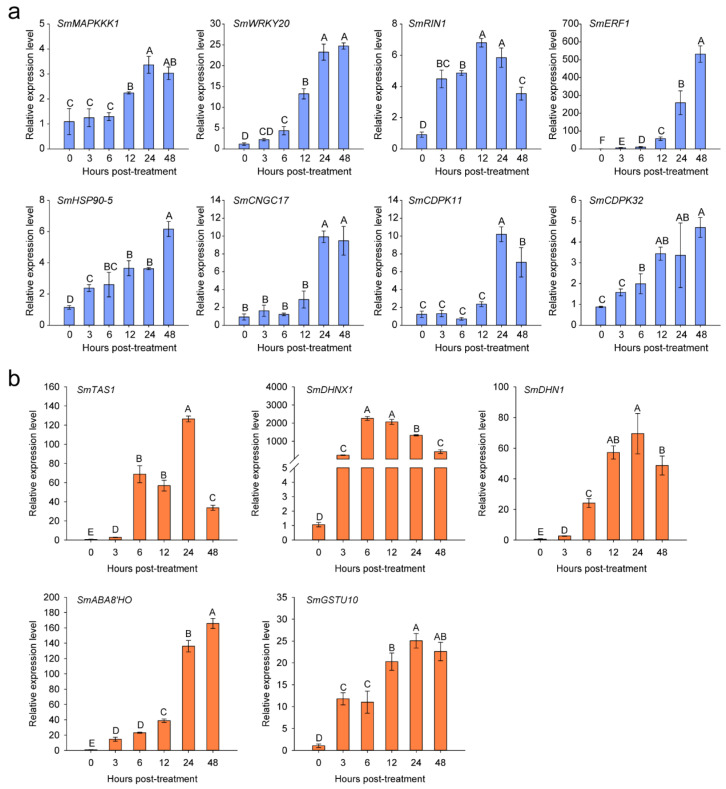
Transcript expression levels of genes associated with the plant–pathogen interaction pathway (**a**) and the ABA signal transduction pathway (**b**) in eggplant roots under salt stress detected by a qPCR assay. Data are means ± standard deviation from four biological replicates. Different capital letters between samples denote significant differences according to one-way ANOVA and Tukey’s test (*p* < 0.01).

**Figure 5 plants-11-02205-f005:**
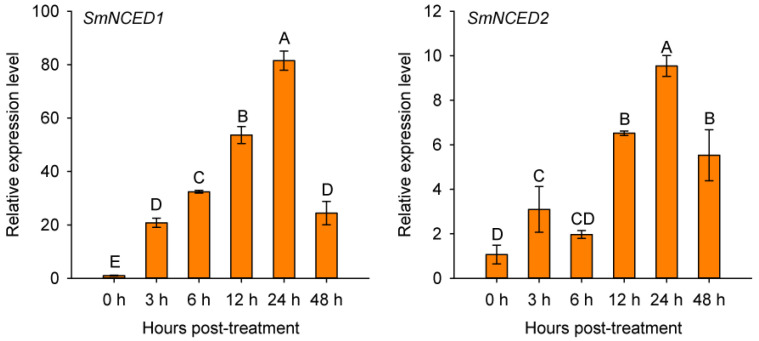
Analysis of the relative transcript expression levels of marker genes related to salt stress responses in eggplant roots under salt stress inferred by a qPCR assay. Data are means ± standard deviation from four biological replicates. Different capital letters between samples denote significant differences according to one-way ANOVA and Tukey’s test (*p* < 0.01).

**Figure 6 plants-11-02205-f006:**
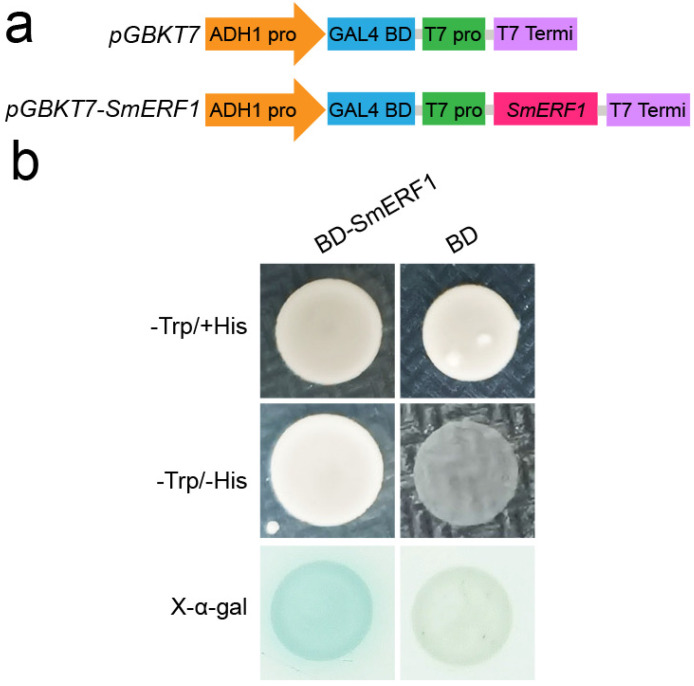
Transcriptional activation activity assay of *SmERF1* using the yeast GAL4 system. (**a**) Diagram of pGBKY7 and the recombinant pGBKT7-*SmERF1* vector. (**b**) Growth of yeast cells transformed with the pGBKT7 empty vector or recombinant pGBKT7−*SmERF1* on a medium supplemented with X−α−Gal but lacking His and Trp. Blue color indicates the activation of the transcriptional activity. BD, binding domain.

**Figure 7 plants-11-02205-f007:**
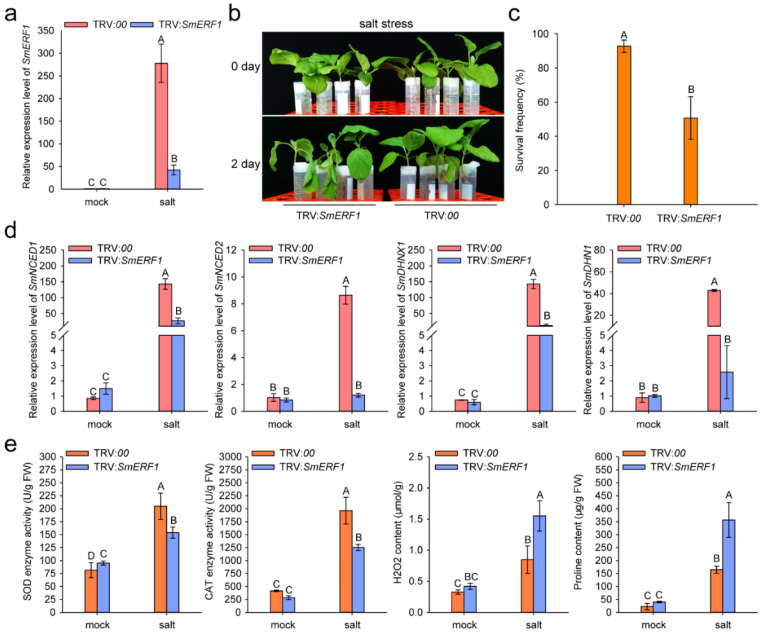
Effects of *SmERF1* silencing on eggplant responses to salt stress (TRV:*SmERF1*, *SmERF1*-silenced; TRV:*00*, control). (**a**) Silencing efficiency of *SmERF1* in plants under a salt stress base on a qPCR assay. (**b**) Phenotype of *SmERF1*-silenced and control plants challenged with salt stress at 2 days post-treatment. (**c**) Survival frequencies of *SmERF1*-silenced and control plants subjected to salt stress at 2 days post-treatment. (**d**) Relative transcript expression levels of the salt stress defense-related marker genes *SmCNGC1*, *SmCNGC2*, *SmDHNX1*, and *SmDHN1* in *SmERF1*-silenced and control plants under salt stress at 48 h post-treatment. (**e**) Proline and H_2_O_2_ content and catalase (CAT) and superoxide dismutase (SOD) activity in *SmERF1*-silenced and control plant roots under salt stress at 48 h post-treatment. In (**a**,**c**–**e**), data are the means ± standard deviation from four biological replicates. Different capital letters between samples denote significant differences according to the one-way ANOVA test (*p* < 0.01).

**Figure 8 plants-11-02205-f008:**
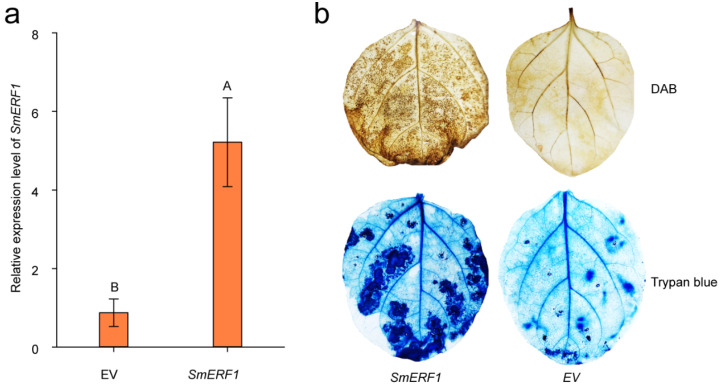
Transient overexpression of *SmERF1* in eggplant leaves. (**a**) qPCR-based detection of *SmERF1* transcript expression levels in leaves with transiently overexpressed *SmERF1* and the empty vector (EV) at 48 h post-infiltration. (**b**) DAB and trypan blue staining of leaves infiltrated with *Agrobacterium tumefaciens* GV3101 cells transformed with *35S:SmERF1-GFP* or *35S:GFP* constructs at 3 days post-infiltration. Data are the means ± standard deviation from four biological replicates. Different capital letters between samples denote significant differences according to Student’s test (*p* < 0.01).

**Figure 9 plants-11-02205-f009:**
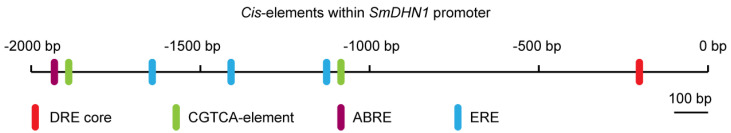
Diagram of *cis*−elements within the *SmDHN1* promoter.

## Data Availability

Not applicable.

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
