# Peer review of "Transcriptome Analysis of Eggplant under Salt Stress: AP2/ERF Transcription Factor SmERF1 Acts as a Positive Regulator of Salt Stress"

_plants, 2022, doi:10.3390/plants11172205_

Round 1
Reviewer 1 Report
The authors did a transcriptome analysis on root tissues of Eggplant to investigate the molecular mechanisms of plant response to salt stress. They showed that SmERF1, a member of AP2/ERF transcription factor family belongs to the plant-pathogen interaction pathway and plays a positive regulatory role in response to salt stress.
It is unfortunate to see how poorly they analyzed the transcriptome. There are too many issues I would like to mention, but I am afraid I cannot write down all of them. However, I am just addressing here a few of them.
1. Figure 1b is completely mislabeled. They mentioned that red dots are the downregulated ones, however I see them on the right side of the figure, which should be upregulated ones. I wonder if they have analyzed it in a correct way! Also, the title on top of the figure says Salt-0 h vs Salt-24 h, which sounds like they sorted the genes based on upregulation and downregulation in Salt-0 h vs Salt-24 h. This should be other way around, I think.
2. In figure 1C, they mentioned the values are transcript expression levels of DEGs. However, transcript expression levels or log2FoldChnage cannot be in a range of 2 to -2, should be bigger range. I think they have used Z-score or something else which they did not mention.
Besides what was their criteria for significant DEGs?
3. They mentioned, “Within biological processes, upregulated DEGs were significantly enriched in 126 pathways, including response to oxygen-containing compounds (GO:1901700), response to acid chemicals (GO:0001101), response to inorganic substances (GO:0010035), and response to water (GO:0009415) (Figure 2a)”.
I don’t see any of them in figure 2a. Again, what was their criteria for significance?
4. Based on the above statement, they said, “This indicated the role of ABA in plant response to salt treatment and suggested that salt upregulates the expression of the genes involved in these pathways."
How these pathways indicate the role of ABA? They are not directly related to ABA biosynthesis or regulation pathways. Why have they selected only these pathways out of 1000+ pathways that they found in GO analysis? There is no explanation.
5. “The results of KEGG analyses showed that salt stress treatment upregulates the transcript expression levels of genes related to plant-pathogen interaction and plant hormone signal transduction (Figure 3a). Therefore, we selected eight plant-pathogen interaction-related genes, namely SmMAPKKK1, SmWRKY20, SmRIN1, SmERF1, SmHSP90-5, SmCNGC17, SmCDPK11, and SmCDPK32, and five ABA signal transduction-related genes, that is ABA and environmental stress-inducible protein TAS14 (SmTAS14), a late embryogenesis abundant protein family member Dehydrin (SmDHN1) [36], Dehydrin Xero 1 (SmDHNX1) regulated by ABA signaling [37], abscisic acid 8'-hydroxylase 3 (SmABA8’HO3), and glutathione S-transferase U10 (SmGSTU10), and detected their transcript expression levels in eggplant roots under salt stress by quantitative real-time PCR (qPCR) assay.”
How did they come up with this list of genes for qPCR? Was it based on the previous studies (which I guess was the case) or did they found these genes in the transcriptome data? If they found it in the transcriptome, they need to list these genes with their ID, name, log2FoldChange and all in a table.
6. All the supplementary tables are poorly prepared. They do not even contain the content as they mentioned in the manuscript.
7. How did they come up with the idea of selecting only SmERF1 for downstream analysis?
They mentioned “KEGG analysis showed that SmERF1 belongs to the plant-pathogen interaction pathway, and its relative transcript expression level was significantly upregulated by salt stress treatment. This suggests that SmERF1 may play a role in eggplant response to salt stress or pathogen attack.”
They checked the expression of other genes with qPCR, then they picked this one only. Why? Was it based on the previous studies? Because this is a well-known abiotic stress responsive TF. If that is the case, they did not need to do transcriptome analysis. They could just select SmERF1, saying they chose this one based on the previous studies. Transcriptome analysis is redundant in that case.
8. They did sequence analysis of SmERF1, which is also unnecessary.
9. They checked tissue-specific expression of SmERF1 and found the highest expression was in leaf. However, they did the transcriptome analysis on root. This does not make sense. Either they could do the transcriptome analysis on leaf tissues, or they did not even need to check the tissue specific expression for this gene. It did not add any significance on the paper.
10. This goes for localization assay. SmERF1 is a transcription factor, hence it would be localized in the nuclei. Showing this was irrelevant and unnecessary.
11. I appreciate they showed silencing of SmERF1 enhanced susceptibility of Eggplants under salt stress. They also showed downregulation of downstream genes under salt stress. This was a good experiment.
However, they should log transform the y-axis label (relative expression level) to make it more consistent and better look.
There are definitely many more issues that I could mention.
I see they used many sophisticated, state-of-the-art techniques, but without a proper planning of what they want to accomplish with those experiments.
I suggest they should take time, analyze the data again, think about how they should present and maintain a logical flow throughout the manuscript. They can follow some of the transcriptome analysis papers to see how they could improve it.
Reviewer 2 Report
In this study, the authors demonstrated that the AP2/ERF in eggplant is involved in the salt stress response by transcriptome and analysis of transgenic eggplant using virus inducing system. The authors suggested possible mechanisms underlying salt stress tolerance. Overall, the study is of worth; but the results and discussion need to be improved before further consideration.
In Title
I think that AP2/ERF1 from eggplants is a negative regulator for salt stress because the silencing of ERF gene shows resistance to salt stress.
In Abstract
Why are authors focusing on ERF1? And How did you find it?
In Abstract
What are DHN and NCED? You shouldn't mention the abbreviation here. If so, please indicate the official name.
In Abstract “These findings suggest that SmERF1 acts as a positive regulator of eggplant response to salt stress. Hence, our results suggest that plant disease-induced AP2/ERF transcription factors play a vital role in the response to salt stress.”
It is an excessive description. The author is investigating that AP2 / ERF1 is involved only in salt stress from eggplant. So please discuss along with the existing results.
Line 123
Here simply describes the results. Also, tell us what does dataset reflects?
Please describe the function of ABA in root briefly.
Line 126
Here simply describes the results. Also, tell us what does dataset reflects?
Line 166
Why did the authors focus on ERF1?
Line168
Please explain based on the author’s results. The pathogen is not related to this manuscript.
Line264-268
Please move to the M&M section.
Line 272
What is TRV:00? Is this an empty vector?
Line 278-281
numerical data is missing. The results regarding up/down-regulation must be discussed by %/folds or numerical data.
In Figures 10 and 11
Authors are kindly asked to write a general statement about the sampling (when was performed, which leaves, number of biological replicates, number of leaves/replicate)
On Figure 11
Multiple comparison test is not suitable for comparison between two data.
Round 2
Reviewer 1 Report
1. They mentioned, The upregulated DEGs were significantly assigned to 127 pathways in biological processes (line 113-114).
Then again in line 125, They said, Within biological processes, upregulated DEGs were significantly enriched in 126 pathways. Once 127 and then 126!
2. In KEGG analysis, they mentioned, The 14 up-enrichment pathways and I see these 14 up enriched pathways in S3. But in figure 3a, there are 20 pathways!
3. I am still not convince how they chose the ABA related pathways to work elaborately based on their Transcriptome data. The related BPs they mentioned were not even the pathways with highest number of genes or with lowest p-value. Selecting that was a wild guess, it does not sound scientifically logical.
These are just few examples. I can mention many more. This study is still poorly analyzed, organized, and presented.
There are many interesting aspects in their transcriptome data. They should spend an ample amount of time, find some interesting features and work on that.
They can re-write this paper without the transcriptome data, and try to publish somewhere.
Reviewer 2 Report
Few revisions are necessary for publication in this journal.
Line73 please revise "ENCD" to NCED
Figure 7 Student's test is not inappropriate for multiple comparison. Before revised, authors used "the one-way ANOVA and Tukey’s test". I think previous one is fine.
Figure 8 I think Student's test is enough for showing difference.
Figure 7a Expression data of SmERF from mock and salt treatments was changed between before and after revisions. Previous one is fine.
Regarding to "Point 2: Why are authors focusing on ERF1? And How did you find it?", i can not find additional sentence on manuscript. Probably, revised version is a preprint version with lines, so the number of lines may have changed. Could you tell me correct line?
